# A Study of the Agricultural Water Supply at the Hoengseong Dam Based on the Hydrological Condition of the Basin

**Deokhwan Kim**

Department of Hydro Science and Engineering Research, Korea Institute of Civil Engineering and Building Technology (KICT), Goyang 10223, Korea; kimdeokhwan@kict.re.kr

**Abstract:** Since South Korea relies on dams and agricultural reservoirs for approximately 45% of its annual agricultural water usage, the supply control of agricultural water based on its usage amount is essential for effective water resources management. The objective of the study, therefore, is to suggest an alternative to the agricultural water supply from the Hoengseong Dam using the threshold curve of usage record that was suggested in the previous study. The characteristics of agricultural water usage and Usage Rate compared to the Permission amount (URP) threshold curve of the Seom River basin, which is defined as the thresholds of the maximum usage rate of agricultural water in each hydrological condition, were derived and analyzed using the historical record of runoff and agricultural water usage. The agricultural water supply of the Hoengseong Dam was simulated from 2006 to 2021 according to the URP threshold curve. As a result, it was found that the agricultural water usage rate of the Seom River Basin can be up to 106.5% even with the average hydrological condition compared to normal years. It was also shown that about 1.7 million m$^3$ of agricultural water could be stored by applying the URP threshold curve to the agricultural water supply of the Hoengseong Dam annually.

**Keywords:** estimation of agricultural water usage; threshold curve; agricultural water supply of multi-purpose dam

## 1. Introduction

Of South Korea's annual agricultural water consumption of 15.2 billion m$^3$ [1,2], the amount supplied through irrigation facilities is 10.1 billion m$^3$ per year, of which 6.8 billion m$^3$ is supplied from multi-purpose dams and agricultural reservoirs [3]. As approximately 45% of annual agricultural water consumption relies on dams and agricultural reservoirs [2], it is important to efficiently manage agricultural water in dams and agricultural reservoirs. Supply management, therefore, is needed concerning the amount of agricultural water used, but it is still insufficient for many reasons.

In general, the agricultural water consumption or demand for supply management has been estimated using two methods: the net water consumption and the water requirement [4,5]. Water requirement is defined as the gross usage of water regarding water conveyance and loss of supplied water, in addition to the quantity required to maximize crop production [6]. Net water consumption is estimated using increased or decreased water resources in a basin due to water used in arable areas. The amount of change is defined as the consumed amount of usage [7,8]. The reason agricultural water consumption is estimated by applying two methods is that consumption changes with artificial factors, such as water management practices as well as regional features including climate, soil, and precipitation, along with seasonal features and time variation, all working in complexity [8–10]. For this reason, the Comprehensive Long-term Water Resource Plan [11] suggested a difference of approximately 38% as it estimated the required amount of water as 1200 mm and the net water consumption as 750 mm based on irrigated paddy fields.

In a study by [12], the required amount of water was estimated as 1200 mm and the net water consumption as 550–660 mm for the Jeju island of South Korea, which shows big differences by who estimated.

Therefore, agricultural water consumption often considers both the required water amount and the net water consumption, but it is difficult to estimate using quantitative models or calculation methods due to big differences that occur depending on the operators of agricultural water facilities or the actual water usage patterns of farmers [13–16]. It makes it difficult to figure out the "amount of water supply and intake" when operating agricultural irrigation facilities such as multi-purpose dams, agricultural reservoirs, and pumping stations, thereby exacerbating the difficulty of water management. In particular, the burden of agricultural irrigation facilities increases as the facilities supply agricultural water, which was supposed to be supplied by groundwater to soil or ground wells when hydrological conditions worsen, such as drought [17–19]. Therefore, countermeasures have continued, such as establishing a preliminary water supply plan or hazard map for drought [20], and many studies contributed to estimating agricultural water consumption. Studies of the water balance of agricultural land concerning meteorological and hydrological conditions [21], and on the agricultural water demand for water resource management of lakes [22], are representative. In addition, the following studies were conducted on: demand estimation for agricultural water in consideration of irrigated crops and regions using satellite images and GIS(Geographic Information System) data [23]; and the estimation of the spatial distribution of agricultural water using Terra MODIS and Landsat 5 TM satellite images [24]. A study on the overall trend and changes in agricultural water requirements using meteorological factors was conducted for Bangladesh [25]. For the regions of South Korea, which is the study area of this study, many studies that contribute to estimating agricultural water supply were also conducted. A supply amount of agricultural water was estimated through an inverse calculation using changes in the water level of agricultural reservoirs [26]. An inverse relationship between the antecedent precipitation and the paddy water usage was suggested [1,27]. The preceding study suggested estimating agricultural water usage based on the relationship between basin run-offs and historical water usage, assuming that the actual water intake record could represent the real amount of water usage. In addition, several studies that tried to estimate actual water usage for paddy fields were conducted. The amount of water intake in an agricultural pumping station, which doesn't have a flow meter, was inversely calculated from the electricity usage records [28], and the methodology and system were suggested to determine agricultural water usage at a nationwide level according to the results of a nationwide hydrographic survey [29]. However, most of them are focused on the amount of demand, and it is still insufficient for application studies, which estimate and predict agricultural water used for the operation of the dams and agricultural reservoirs.

As follow-up research of [1], the objective of this study is to suggest a practical methodology to estimate the actual agricultural water supply of the Hoengseong dams in consideration of the hydrological situations of a basin and examine its applicability. The previous study [1] suggests a threshold curve concept that describes the maximum usage of agricultural water in each hydrological condition. However, it needs to complicate analysis to apply in actual practice because it is based on the maximum usage. Therefore, the modification and improvement for practical application is the major objective of the study. To achieve this, the historical records of agricultural water intake, dam operation, and runoff of the Seom River basin were obtained. It analyzed the characteristics of agricultural water usage in the Seom River basin, and the "Usage Rate compared to Permission amount (URP)" threshold curve for agricultural water usage was suggested using the concept in the previous study. It simulated and applied the URP threshold curve to the actual supply of agricultural water of the Hoengseng Dam and verified the applicability of the suggested method. It also compared the permission amount of agricultural water with the maximum value of the threshold curve and classified the characteristics of agricultural water usage into three types.

## 2. Theoretical Background

### 2.1. Study Material

The study area is the Hoengseong Dam and the Seom River basin. The Seom River basin is the first tributary of the Han River located in the central part of the Korean peninsula, with a basin area of 1490.1 km² and a river length of 100.6 km, accounting for approximately 4.2% of the entire Han River. The Hoengseong Dam is a medium-sized multi-purpose dam located upstream of the Seom River basin, with a basin area of 209.0 km², a total storage capacity of 86.9 million m³, a flood control capacity of 9.5 million m³, and an annual water supply of 119.5 million m³(Figure 1.). The dam plays a major role in water use in the Seom River basin, specifically, domestic & industrial water supplying the same amount throughout the year, and agricultural water supplied only during the irrigation period (April to October) [30].

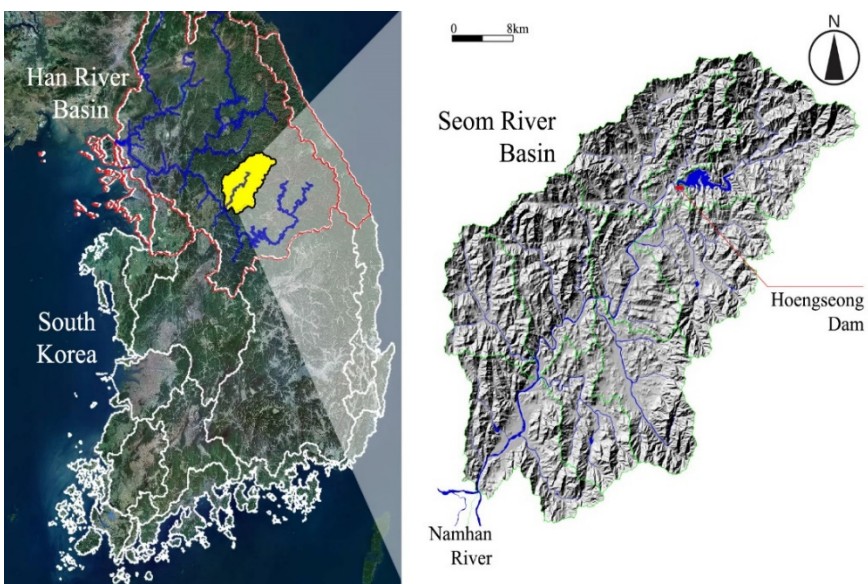

**Figure 1.** Study Area and basin.; red- and whit-line indicate the Han River and other major basins in South Korea respectively, and yellow area indicate the Seom River basin.

The Seom River basin is considered a suitable area for studying the agricultural water supply. The agricultural area of the basin is 416.2 km², and the rice field is 151.5 km², corresponding to 27.9% and 10.2% of the total, respectively [11]. The pattern of agricultural water usage is akin to the water demand for rice, a major crop of the Korean peninsula [19]. 37% of the totalagricultural water is used during the tillering stage (April to June) in the Seom River basin [31]. The actual demand for agricultural water would vary depending on the stage and hydrological conditions. The current method that supplies the same amount throughout the entire period will supply an unnecessary amount. Therefore, the agricultural water supply of the Hoengseong Dam could be improved and tested for more efficiency(Figure 1).

To conduct this study, hydrological records of the basin and the dam were essential, and the period is from 2006, when reliable data began to be collected, to 2021. The flow records of Wonju (Jijeong Bridge) were used as the representative runoff of the Seom River basin since it is an area located in the downstream area of the basin and a quality control site to conduct low-flow forecasting. Historical records for 2006–2020 were collected from the authorized Korea Annual Hydrological Report [32], and records for 2021 were collected through the Water Resources Management Information System [33]. It is required to report if 8000 m³ or more of agricultural water is used per day under the River Act. Since 18 agricultural pumping plants use 98,000 m³ per day in the Seom River basin, the agricultural water usage records were acquired through the Han River Flood Office of the Ministry of Environment [34]. Hydrological records of the dam were collected from the

information page of the Korea Water Resources Corporation [30]. All obtained records were checked for outliers, and erroneous values were removed.

*2.2. Threshold Curve of Agricultural Water Usage Ratio Compared to Permission Amount*

Ref. [1] Suggested the 'Water Usage rate compared to Maximum usage amount (WUM)' and its threshold curve method for estimating water demand. It estimates the maximum usage amount using the percent of normal year runoff for the last 2 months based on the correlation between basin runoff and agricultural water usage that is assumed as the 'threshold curve'. The most important assumption suggested in [1] is that "the maximum usage of agricultural water is inversely proportional to the hydrological conditions of the basin, and increases during the dry period and decreases during the wet period". Considering the situation in South Korea, where the limit on river water usage is set and managed to prohibit the usage of more than the permitted amount, it seems that the maximum usage of agricultural water in each hydrological condition (threshold curve in [1]) can be used as an alternative to the fixed permission amount on agricultural water management. In particular, Article 50 of the River Act and Article 8 of the Enforcement rule of the same Act stipulate that the unused volume of water usage permission may be adjusted (or reduced) if the annual average usage is no more than 40% of the permitted volume; the average usage over the recent three years is no less than 60% of the permitted volume; or the average usage over the recent five years is no less than 80% of the permitted volume. In the case of residential and industrial water, this adjustment may be reasonable because a certain amount of water is consistently collected regardless of seasonal changes in many cases, but it is difficult to apply the same standard to agricultural water. As agricultural water is used only during the irrigation period (April to October in South Korea), it is difficult to estimate the annual average usage rate, and the demand tends to increase during droughts [16,18]. In the case of supply management or adjustment based on the average usage rate, it has been suggested that there may be a shortage of agricultural water in future droughts. Therefore, even if it is necessary to adjust the amount of permitted agricultural water, it is difficult to amicably reach an agreement with operators of agricultural pumping plants due to a reason such as "securing capacity in preparation for the occurrence of low-flow (drought)".

It, however, appears possible to make quantitative judgments and manage supply if the threshold curve of agricultural water usage suggested in the study of [1] is applied because the threshold usage (or maximum usage) of agricultural water during droughts is already taken into consideration. Since the study showed that the basin runoff has a certain correlation with the maximum agricultural water usage, a threshold curve can be derived by applying the correlation to the representative basin runoff and the agricultural water Usage Rate compared to the Permission amount (%). As the threshold curve represents the "maximum usage rate compared to Permission amount," it can also be regarded as a usage rate indicating there is no problem with agricultural water use even during droughts. Such threshold curve of the "Usage Rate compared to Permission amount" (hereinafter referred to as "URP") can be derived through the following three simple steps: (See Figure 2).

1. Find a duration ($n$-month) with the highest correlation with the maximum usage rate by diagraming the agricultural water URP value and the basin runoffs by cumulative period ($n$ = 1, 2, 3…, $n$-month).
2. Draw a scatter diagram of the agricultural water URP value and the basin runoffs for a selective cumulative period.
3. Estimate the quantile regression curve with 95% (non-exceedance probability) or the maximum value of the interval in the scatter diagram.

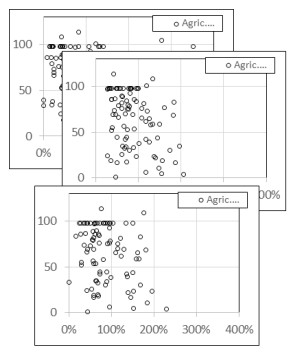
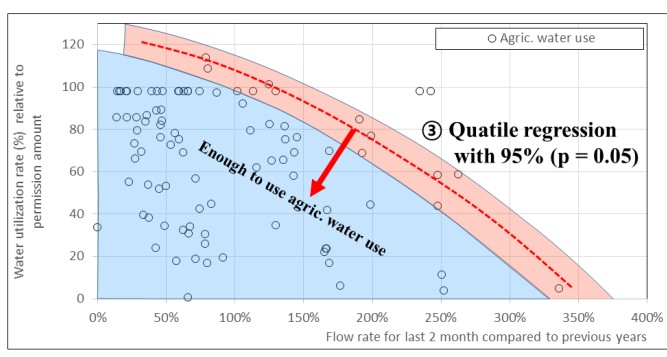

**(a)** Find suitable 'n'-month

**(b)** Scatter diagram 'agric. water usage' vs 'n-month flow rate'

**Figure 2.** Diagram to derive the threshold curve of URP.

Figure 2 describes the process of deriving the regression curve, which consists of a 95% percentile of agricultural water usage in each hydrological condition. Accordingly, the regression curve can be considered a threshold curve of 95% or more usage rate of agricultural water intake (or used agricultural water) under each specific hydrological situation. It is possible to ensure supply reliability of at least 95% for the entire agricultural water usage, even in drought. It corresponds to a 20-year return period which is the target frequency of drought countermeasure to droughts in the multi-purpose dam. Therefore, it is expected that there will be no difficulty in supplying agricultural water at a level that meets the supply reliability of multi-purpose dams.

## 3. Application and Results

### 3.1. Characteristics of Agricultural Water Usage in the Seom River Basin and the URP Threshold Curve

As the arable area in the Seom River basin takes up 416.2 km², accounting for 17.8% of the whole basin area, the agricultural water usage is relatively high, and a significant amount of agricultural water relies on the Hoengseong Dam. The dam supplies 15.8 million m³ of agricultural water annually, but complaints about a shortage of agricultural water occur periodically depending on hydrological conditions. And it was reported that the complaints occurred mainly at the beginning of the irrigation season [30]. Considering that 37% of the total agricultural water is used during the tillering stage (April to June) in the Seom River basin [31], first, the actual usage amount of agricultural water in the basin may be larger than the permitted amount; and, second, it can be considered that the agricultural water consumption is concentrated during the tillering stage from the end of April to the end of June and is greatly affected by hydrological conditions of the basin. Figures 3 and 4 show the analysis results of the Seom River records for the last 15 years.

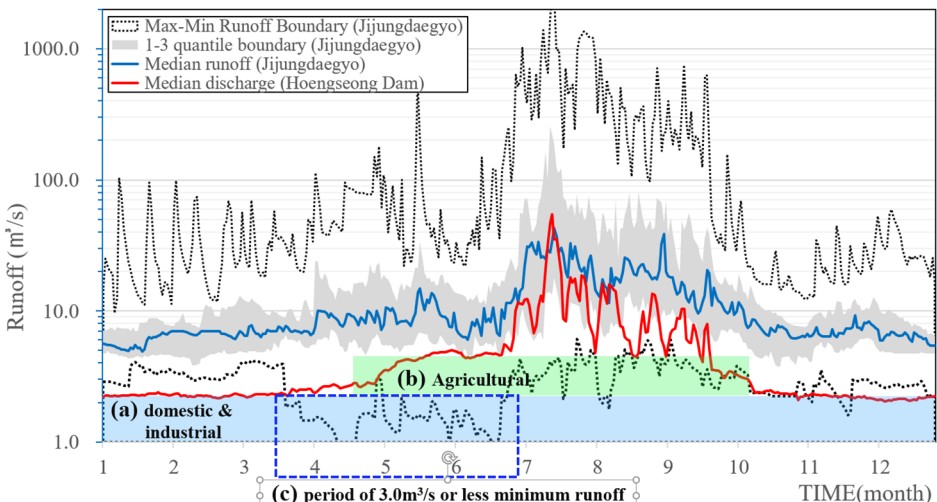

**Figure 3.** Flow diagram of Seom River basin for last 15 years; blue line indicates median runoff rate, red line indicates median discharge of the Hoengseong dam, dot-line indicates maximum to minimum runoff boundary, and gray-colored area indicates 1 to 3 quartile area of runoff, also (**a**) and (**b**) indicate discharged amount for domestic & industrial (blue box) and agricultural water (green-box) in the Hoengseong dam, (**c**) indicate the period of 3.0 m³/s or less minimum flow (blue dot-line box).

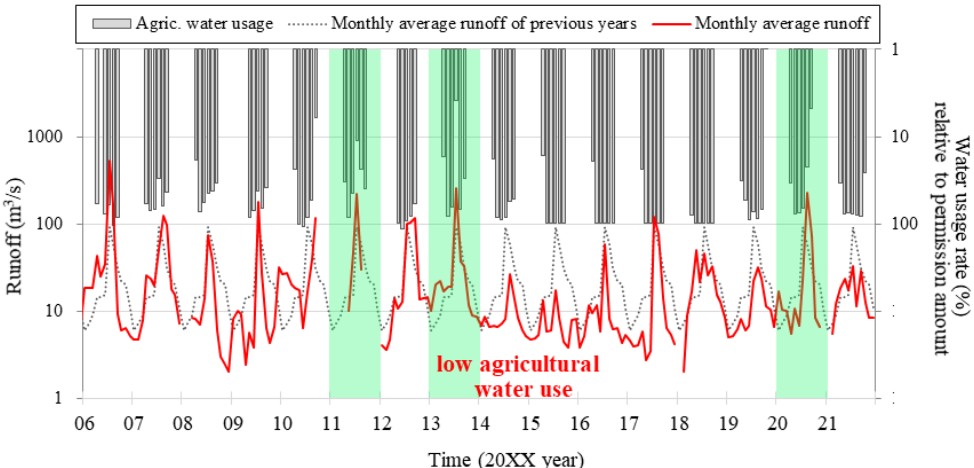

**Figure 4.** Time series of monthly runoff rate and agricultural water usage rate on Seom River basin for last 15 years; red line indicates monthly average runoff, dot-line indicates monthly average runoff of for previous 15 years, gray box indicates the usage percentage of agricultural water relative to permission amount.

The Seom River basin has 16 agricultural reservoirs with a 12.1 million m³ capacity. Most of them supply agricultural water using the pipeline system and the Hoengseong Dam located in the upper part of the basin. Therefore, the basin shows similar runoff behavior to a non-regulated basin. The runoff condition in a non-regulated basin of South Korea is lowest in winter (from December to February), and it gradually increases from March when spring begins, showing the highest flow rate in flood season [35]. In Figure 3, the runoff in the Seom River basin is maintained at a rate of 5 to 12 m³/s throughout the year, and its minimum rate is about 3.0 m³/s. Therefore, it can be considered that the runoff rate of the basin is no less than 3.0 m³/s. However, the runoff condition spreads widely below the first quartile of the runoff (blue dot-line box in Figure 3), particularly from April to June. The minimum runoff is found to be lower than 3.0 m³/s. This means that it may appear below the average minimum runoff rate of 3.0 m³/s during the relevant period, despite that the discharge amount of the Hoengseong Dam was increased by 1.5 m³/s for

agricultural water use. The runoff in spring is higher than in winter, and it is also suggested that the Seom River basin has a higher natural runoff in spring than in winter [35]. Accordingly, the result in Figure 3 is likely due to water use. Since the usage of residential and industrial water does not vary significantly with the change of seasons [35], the change in water usage is mainly due to agricultural water. Therefore, the decrease in the runoff during the tillering stage shown in Figure 3 is believed to be due to the impacts of the use of agricultural water, and it is considered reasonable that the agricultural water usage in the Seom River basin can be higher than the permitted amount.

In addition, it is found that the agricultural water usage rate changes depending on the hydrological conditions of the Seom River basin in Figure 4, a diagram of the agricultural water URP and the average monthly runoff rate from 2006 to 2021. In particular, the overall agricultural water usage rate decreases when the hydrological condition of the basin becomes a wet condition, such as in 2011, 2013, and 2020, and the agricultural water usage rate steadily decreases toward the latter part of the irrigation period (green-box in Figure 4). Therefore, the second assumption that "agricultural water use is concentrated in the tillering stage from the end of April to the end of June and is greatly affected by the hydrological conditions of the Seom River basin" is considered appropriate. Therefore, it is consistent with the assumption suggested in the preceding study [1] and Section 2.2. Thus the agricultural water use of the Seom River basin is affected by its hydrological conditions.

Accordingly, the URP threshold curve methodology presented in Section 2.2 was applied in the Seom River basin. In detail, the monthly agricultural water usage records of the Seom River Basin were converted into agricultural water Usage Rates compared to Permission amount (URP). It was diagrammed with the representative basin runoff with the cumulative runoff of the last several months (Figure 5). Two months as selected as an optimal period of cumulative runoff (Figure 5b). This led to the same result as in the preceding study [1], and the reason seems to be that the agricultural water supplier establishes and decides an operation plan linked to the supply amount of the past one to two months and the amount of water stored in the dam or reservoir [34].

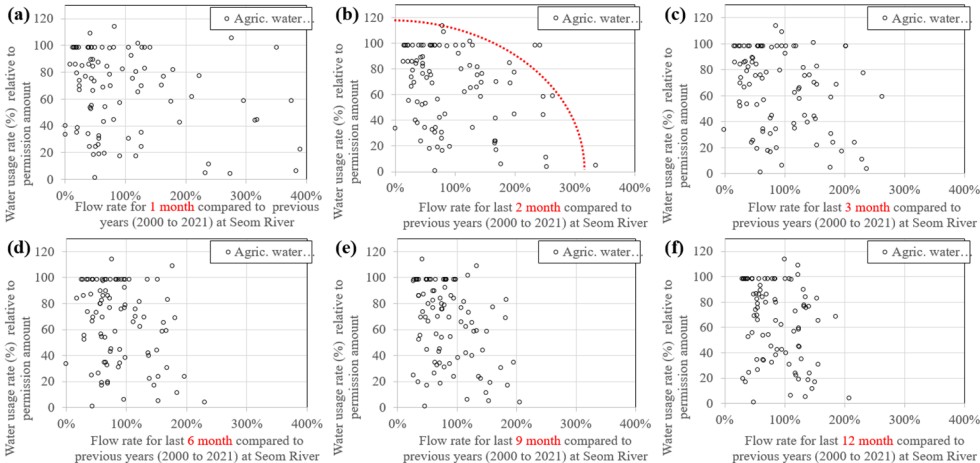

**Figure 5.** Scatter diagram for the percent of normal year for last several months versus water utilization rate compared to permission amount in the Seom River; (**a**) 1-month; (**b**) 2-months and red-dot-line indicate threshold curve of the URP; (**c**) 3-months; (**d**) 6-months; (**e**) 9-months; (**f**) 12-months.

As shown in Section 2.2, the agricultural water URP and the two-month cumulative runoff are diagramed in a scatter diagram, and a threshold curve was derived through quantile regression using the 95% quantile or the maximum value (red point in Figure 6) of which results are shown in Figure 6. However, some agricultural pumping stations in the river water usage data showed 99% usage of the permission amount between 2015 and 2016, as this trend continued even after the irrigation period was finished. Thus, inquiries were made to a relevant institution, Korea Rural Community Corporation (KCRC), and it was decided that the data had different records from the actual amounts. Thus the erroneous records were excluded (green point in Figure 6). If the usage rates of agricultural water for a specific hydrological condition are diagramed through the above methodology, the marginal usage rate (or maximum usage rate) gradually decreases as the hydrological condition improves. Therefore, it could be expressed as a sort of recession curve. Among these, a second-order form of the recession curve shows the best result (Figure 6, $R^2 = 0.961$), the URP threshold curve expressed in the following second-order formula:

$$URP = a \times (PNR_2)^2 + b \times (PNR_2) + c$$
$$= -8.1663 \times (PNR_2)^2 - 6.6513 \times (PNR_2) + 121.3$$

(1)

The URP is the Ratio of agricultural water Usage to Permission amount between 0% and 100%. $PNR_2$ is the ratio of the basin runoff compared to the previous two months, and $a$, $b$, and $c$ are regression parameters of the second-order equation estimated from the threshold curve in Figure 6. The URP curve could differ due to factors such as exponential, nth-order, logarithm, and others.

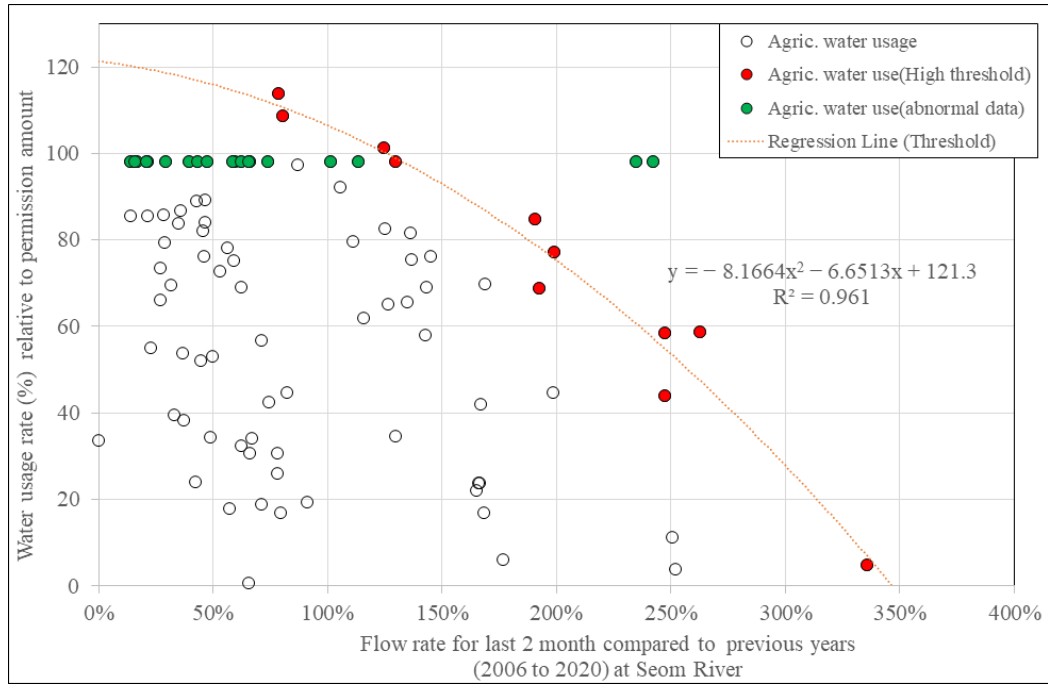

**Figure 6.** Scatter diagram of the percent of normal year runoff and the water usage rate compared to permission amount in the Seom River; red point indicates it has 0.95 or more value in CDF, red dot line is the threshold line of URP, and green point indicates data treated as abnormal records.

Using the URP threshold curve in Equation (1), the usage of agricultural water can be estimated in consideration of the hydrological conditions of the Seom River basin. For example, if the flow rate of the river for the last two months is 100% compared to the previous year, the URP is 106.5%. Therefore, agricultural water in the Seom River basin can be used up to 106.5%, which is more than 100% of the current agricultural water usage permission, even with a hydrological status similar to the previous year's annual average.

These results are consistent with the assumption of this section that "the agricultural water usage in the Seom River basin may be relatively high". The operators of multi-purpose dams or agricultural reservoirs can easily determine the usage rate of agricultural water and its supply amount using the URP threshold curve.

*3.2. Agricultural Water Supply Simulation According to URP Threshold Curve*

In the previous section, the URP threshold curve of the Seom River basin was derived using the method suggested in [1]. Since the Hoengseong Dam is located in the upper part of the basin and plays an important role in supplying agricultural water to the basin, the URP threshold curve (red-dot line in Figure 6) could be used as an alternative for agricultural water supply management of the dam instead of the fixed amount. It seems possible to efficiently supply agricultural water by applying the usage amount calculated from the URP threshold curve, rather than the current method of supplying a certain fixed amount, to the agricultural water supply of the dam. The Hoengseong Dam supplies 1.2 to 3.0 million m³ of agricultural water monthly from April to November. At present, it supplies agricultural water by discharging all of an amount pre-determined monthly. However, this study simulated the supply of agricultural water by applying the URP threshold curve and examining its results. The agricultural water on the Hoengseong Dam from 2006 to 2021 was simulated by supplying the usage amount according to the URP critical curve and stockpiling the remaining agricultural water compared to the existing quantitative supply method. Adjustments to the agricultural water supply monthly may lead to smaller effects or slow responses to changes in hydrological conditions. In the simulation, therefore, the supply amount was changed on a weekly scale according to the URP threshold curve. A maximum of 100% was supplied even if the usage ratio of the agricultural water exceeded 100%.

Figure 7 shows the simulation result that agricultural water can be operated efficiently depending on hydrological situations. This means that the surplus can be used for river management as there is a period in which the agricultural water URP is less than 100% due to changes in hydrological conditions. In particular, it was possible to adjust the supply of agricultural water every year, except in 2015–2017, 2009, and 2019, when severe droughts occurred in the Han River system [36]. Since such supply adjustments do not cause trouble with agricultural water use, it seems to apply to the operation of multi-purpose dams.

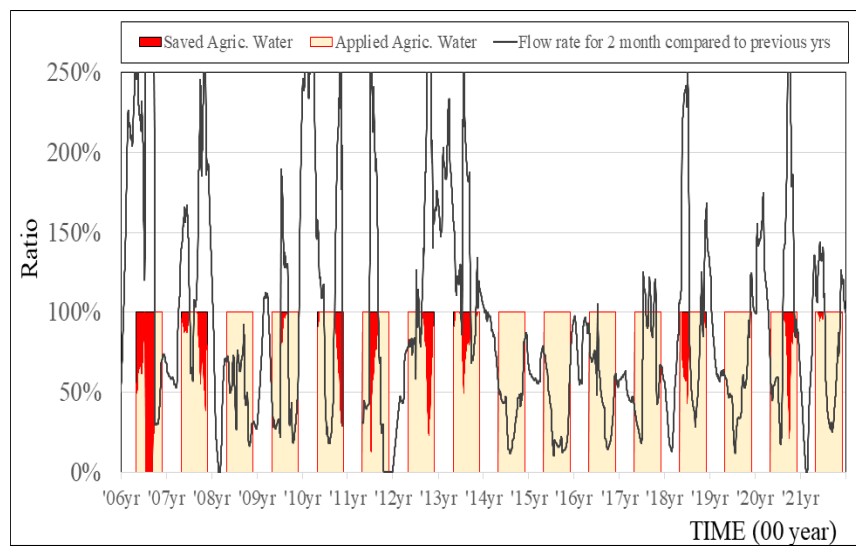

**Figure 7.** Time series plot for agricultural water supply based on the URP threshold curve; red bar indicates the amount of agricultural water saved.

### 3.3. Results and Considerations

This study derived the URP threshold curve of the Seom River basin by modifying the Water Usage rate compared to the Maximum usage amount (WUM) suggested in the study [1] and showed applicability to the agricultural water supply of the Hoengseong Dam. As a result, the total supply amount is 235 million m³ if agricultural water is supplied weekly by applying the URP threshold curve, therefore an additional amount of 18.2 million m³ can be utilized for agricultural or other purposes, and the average annual usable amount is approximately 1.7 million m³ (See Figure 8).

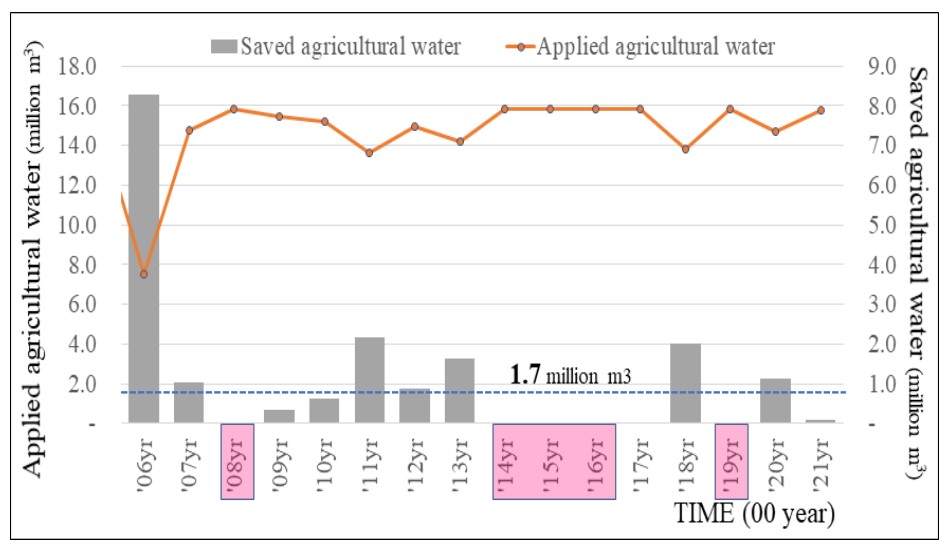

**Figure 8.** Annual supplied and saved amount of agricultural water of the Hoengseong Dam; red box indicated the year that can't save agricultural water due to drought.

Out of 15.8 million m³ of the annual agricultural water supplied by the Hoengseong Dam, 1.7 million m³ accounts for approximately 11%. If one-tenth of the annual supply can be flexibly operated, it will contribute to the efficient operation of multi-purpose dams and river management. The previous section shows a period when the river flow rate of the Seom River basin decreases due to excessive use of agricultural water during the tillering stage. In this case, if the stored agricultural water is discharged as the maintenance water for the Seom River, it will be possible to efficiently make up for the deteriorated ecological environments and water quality or the shortage of agricultural water due to a decrease in the river flow rate.

In particular, the water supply based on the URP threshold curve is not part of supply adjustments that limit the agricultural water usage due to reasons such as drought, but is close to a surplus adjustment within a range where there is no problem with agricultural water use. Therefore, we feel there will be a low level of opposition to agricultural water adjustments from farmers or related organizations. In addition, the agricultural water stored through adjustments can be used to supply additional water when there are water shortages during the tillering stage mentioned in the previous section or as the maintenance water for the river.

Another important point is that the URP threshold curve is expected to show differences depending on the usage characteristics of agricultural water in the basin. The URP value of the Seom River basin is over 100% from 125% or less of $PNR_2$. It means that it is possible to use more than the basin's permitted amount of agricultural water usage despite the better-than-average hydrological conditions of the basin compared to the previous year. It implies that the agricultural water usage in the Seom River presented in the previous section may be relatively larger than the permitted amount. Since the current River Act prohibits water intake exceeding the permission amount, it seems necessary to make adjustments or prohibitions for water resource management of the basin. However,

it has often been difficult to adjust the permitted amount using the existing adjustment process of agricultural water because it is needed to secure a capacity to prepare for the occurrence of low-flow (drought) due to an increase in agricultural water usage during droughts. In this case, the URP threshold curve could be an alternative to identify the usage characteristics of agricultural water and make adjustments. The basic concept of the URP threshold curve is that it connects usage thresholds of agricultural water intake (used) in various hydrological situations such as dry and wet seasons. Therefore, it can be seen as a curve that reflects the agricultural water usage at the time of low flow (drought), and it is expected that the curve can be used for analysis to make adjustments to agricultural water supply and permission amounts of a basin or dam. In principle, the ideal case of water resource management would be if the maximum value of the URP threshold curve is exactly the same as the supply/permission amount of agricultural water. However, there may be cases where the maximum value of the URP threshold curve is not consistent with the supply/permission amount for various reasons.

Figure 9 divides the URP threshold curve into three types. The blue line indicates the permission amount of agricultural water use, and the red line indicates the URP threshold curve. Figure 9a is the case where the maximum value of the URP threshold is equal to the permission amount of agricultural water. In terms of water resource management, this is an ideal case as agricultural water is supplied within the existing permitted amount even during drought. As the hydrological condition improves, unused agricultural water can be utilized, and adjustments to the permission amount are not required. However, if the maximum value of the URP threshold curve exceeds the permission amount of agricultural water, as shown in Figure 9b, it means that the agricultural water usage exceeds the permitted amount during drought. Since this is a case with a problem with agricultural water use under the current law, the permission amount of agricultural water must be increased, and the required increase amount can be estimated by calculating the difference between the maximum value of the URP threshold curve and the permitted amount. As shown in Figure 9c, if the maximum value of the URP threshold curve is less than the permitted amount of agricultural water, it means that the amount of the used agricultural water is lower than the permission amount even during drought, and it can be interpreted that there is an unnecessary surplus in the permitted amount. Therefore, it will be possible to lower the permission amount of agricultural water up to the maximum value of the URP threshold curve and utilize the remaining water resources. In addition, since the URP threshold curve can be derived considering hydrological conditions and agricultural water usage, the curve can be applied to individual facilities that use agricultural water and basins. And the curve can be applied to adjustments to the permission amount of facilities using agricultural water under Article 50 of the River Act and Article 8 of the Enforcement Rule of the River Act. Therefore, it is expected that the URP threshold curve methodology for agricultural water suggested in this study can be used by operators of multi-purpose dams or agricultural reservoirs and persons who manage use permits for river water to easily estimate agricultural water usage or make adjustments concerning hydrological situations.

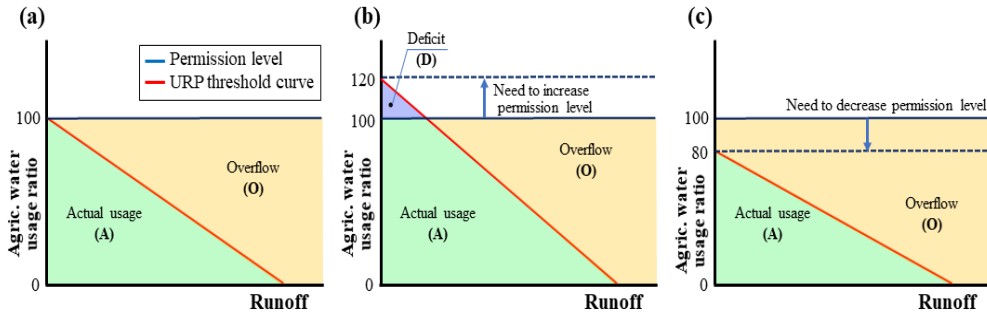

**Figure 9.** Three types of the URP threshold curve; (**a**) maximum value of the URP curve are equal with permission amount, (**b**) maximum value of the URP curve are larger than permission amount, (**c**) maximum value of the URP curve are smaller than permission amount; blue dot-line in (b) and (c) are the permission amount that need to increase or decrease.

A limitation of this study is that the reliability of the URP threshold curve entirely depends on data quality. Since the basic URP concept is to use usage thresholds of agricultural water or intake in a specific hydrological situation, the derived threshold curve can be convincing only when the existing data are reliable while including records on agricultural water usage in various situations from dry to wet conditions. For example, the historical record on agricultural water usage in the Seom River basin was not used for the analysis. It included a uniformly equal record of agricultural water usage as 100% compared to the permitted amount between 2015 and 2016. Another important limitation of the study is that the suggested method belongs to interpolation. Since the demand for agricultural water may be higher in the case of a severe drought that has never occurred before, it could not be used to establish a plan or design for the future. Another limitation is that the suggested method is derived from data from South Korea. The pattern of agricultural water use in South Korea is akin to the water demand for rice which is a major crop on the Korean peninsula. Therefore, another major crop would have its own water demand pattern, meaning that the URP curve could be useless for another major crop. However, the method presented in this study allows easier estimation of agricultural water usage, and it can be utilized for water management of multi-purpose dams and reservoirs.

## 4. Conclusions

As a follow-up study [1], our study suggests a methodology to estimate the actual agricultural water supply of multi-purpose dams considering hydrological situations and analyze its results for future application. To achieve this, the historical records of agricultural water intake, dam operation, and runoff of the Seom River basin from 2006 to 2021 were obtained. It analyzed the characteristics of agricultural water usage in the Seom River basin. Based on records over the past 15 years, it suggested that the agricultural water usage in the basin may be higher than the permitted amount; the water usage is concentrated in the tillering period from the end of April to the end of June, and the usage is greatly affected by hydrological situations. It also derived the URP threshold curve in the Seom River basin, considering its hydrological situation by presenting and applying the "URP threshold curve" using WUM presented in the preceding study. It pointed out that agricultural water in the Seom River basin can be used up to 106.5%, which is more than 100% of the permitted amount of agricultural water, even in a hydrological condition similar to the previous year's annual average. The study verified its adaptability by simulating the agricultural water supply of the Hoengseong Dam using the URP threshold curve. It showed an average of 1.7 million m$^3$ of agricultural water (out of 18.2 million m$^3$ in total) could be stored or operated annually. It also verified the usability of the threshold curve in making adjustments to the permission amount of basins and dams. It classified the URP threshold curve into three types concerning the permitted amount of agricultural water and the maximum value of the curve. It is hoped that the method suggested in this study will be used in water management in the future. The method allows easy estimation of agricultural water usage in terms of water management, and it can be used for supply management. But, as it also needs reliable data and could not be used for planning or designs, the method will have to be used carefully in actual practice.

**Funding:** This work was supported by the Korea Environment Industry & Technology Institute(KEITI) through the Water Management Program for Drought Project, funded by the Korea Ministry of Environment(MOE),(2022003610002).

**Institutional Review Board Statement:** Not applicable.

**Informed Consent Statement:** Not applicable.

**Data Availability Statement:** Not applicable.

**Acknowledgments:** This work was supported by Korea Environment Industry & Technology Institute(KEITI) through the Water Management Program for Drought Project, funded by the Korea Ministry of Environment(MOE),(2022003610002).

**Conflicts of Interest:** The authors declare no conflict of interest.

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
