# Peer review of "A Study of the Agricultural Water Supply at the Hoengseong Dam Based on the Hydrological Condition of the Basin"

_water, doi:10.3390/w14162508_

Round 1
Reviewer 1 Report
This manuscript is appropriate for the Water journal. The paper is providing a unique and powerful approach to contemplate agricultural water supply in multipurpose dams or reservoirs. This paper is well documented. I am impressed with the flow of the entire manuscript. The methodology is well documented. This paper needs some minor grammar corrections but it can be accepted for publication with these minor corrections.
· Line 7: Specify the source
· Line 11: Mentioning the study name or method name will be more meaningful
· Line 25: Remove ‘and’ at the end of sentence
· Line 58: Give some examples.
· Line 183. Correct the word “replies” to relies.
· Line 186: Correct “irritation” to irrigation.
· Line 188. Change to First instead of ”Firstly”
· Line 189 Change “Secondly” to Second
· Line 318: It should be problem not “problems”
· Line 336: Rephrase sentence to “Can also be used”

Author Response
"Please see the attachment."

Reviewer 2 Report
Congratulations to the author for completing a good job.
Q1. My only question is Hoengseong Dam uses water for multiple purposes. Please indicate how much water is being used for. domestic use, industrial water, power generation water, and agricultural water.
Q2. Are the figures presented in Figures 3 and 4 observational data?
If so, please add a description of the data, such as the source of the observation data, in the manuscript.
Q3. Title of chapter 2.1 typo.
Author Response
"Please see the attachment."

Reviewer 3 Report
General comments
The manuscript “A Study on Agricultural Water Supply Method on the Reservoir based on Runoff Condition of the Basin” presents an approach, as a continuation of a previous study, aiming to estimate the actual agricultural water supply of multi-purpose dams taking into account the hydrological regime. The study is based on a set of 15-year data from Seom River basin (Korea), where Hoengseong multi-purpose dam is located.
The topic of the study is interesting, dealing with the sensitive matter of agricultural water supply, under possible conditions of water scarcity (e.g. droughts) and potentially competitive water uses. However, the manuscript has several weak points as presented in detail in the specific comments below. In brief, the goals of the study are not clearly set, while the proposed approach seems to be mostly limited in the specific case, in which several assumptions have been introduced and, consequently, the presented results do not appear to be sufficiently solid in order to be generalized in other cases.
Specific comments
- [Title] A revision of the title is suggested; it is difficult to comprehend the selected phrasing.
- [Abstract] Several facts are unclear in the abstract, e.g. which is the “particular source” (L.7) on which South Korea relies on, or which is the “previous study” (L.11) that is referred to? In any case, the abstract should be thoroughly revised to present briefly the background of the topic that leads to the objectives and goals of the specific study.
- [Section 1] The introductory section presents a general literature review regarding agricultural water use. However, the actual issues which are intended to be investigated, as well as the objectives, goals and novelty aspects of the study are unclear. It is recommended to focus on the specific topic of interest, setting clearly the investigated problem and the intended goals. It seems that the study is a follow-up of a previous research, therefore it must be further emphasised the reasons leading to the need of this follow up and the added value of the current study.
- [Section 2] Following the previous comment, it is not clear why the selected study area is suitable for studying the agricultural water supply. More specifically, it is stated that this is due to the fact that “it uses a relatively large amount of agricultural water due to its high ratio of paddy field to the basin area and relies on the Hoengseong Dam for a significant portion of agricultural water supply” (L.106-108). However, the paddy fields cannot be considered as a characteristic case of typical agricultural water use, due to the high water requirements, therefore the suitability of the area for the study is not evident and must be clearly explained (along with the objectives that should be set in the introduction). Furthermore the “threshold curve method” (L.123) which is mentioned to be used in the previous study should be briefly described, as well as its components (e.g. “the maximum usage of agricultural water” – L.131). It is also noted that the use of this threshold approach (as it is also shown later in the results), does not seem to take into account any of the other water uses, considering that the case refers to multi-purpose dams; please justify/elaborate on this issue.
- Considering that agricultural water consumption seems to be directly related to the development stages of the main crops of the area (as presented in Section 3), additional information on the development stages of the crops for the specific study area should be provided in Section 2.
- [Section 3.1] Please clarify whether the presented runoff amounts (Fig.3) are before or after the regulated discharge due to the existing reservoir. This is important regarding the assumption that the decrease of runoff is due to agricultural consumption (L.209-212), which is the basis for the further presented case. It is noted, that there are several references mentioning similar assumptions and results presented in “the preceding study” (Kwan, 2021), though the international readership is rather difficult to follow these statements due to the fact that the aforementioned study is written in Korean. Additional details should be provided regarding the base of the assumptions and the results which are mentioned regarding the study of Kwan (2021).
- [Fig. 4] Please clarify what is the ‘monthly average runoff of previous years’ and why is it presented in the graph. Also, the x-axis labels (‘06yr, ‘07yr, etc.) should be changed (i.e. 2006, 2007, etc.). The latter applies also for Fig. 6 & 7.
- [Equations 1 & 2] These equations are poorly justified and not supported by any relevant references, while the explanation of the presented terms is also rather weak. For instance, it is not clear which are the “characteristics of the basin” on which the “critical constants” a, b and c are based on and why (L.248-250).
- [Section 3.2] The presentation of the approach in the first paragraph of the section should be further elaborated. Furthermore, the statement that “severe droughts occurred in the Han River system” (L.292) is unsupported; considering that one of the is intended goals of the study is the approach to be applicable under drought conditions, it would be expected to justify and quantify the severity of droughts in the period of study, using some typical drought assessment approach (e.g. appropriate hydrological and/or agricultural drought indices), that could also verify the performance of the approach under specific drought severity cases. Finally, it is not understood which exactly are the “supply adjustments” (L.293) which do not have impacts on agricultural water used, leading to the conclusion that these can be applicable to multi-purpose dams; please elaborate/justify.
- The use of the language requires improvement as it is quite difficult to follow smoothly the flow of the text (especially in the first sections); an edit by a native English speaker is advised.
Author Response
"Please see the attachment."

Round 2
Reviewer 3 Report
The revised version of the manuscript includes improvements, many issues have been sufficiently addressed according to the previously raised comments, while clarifications have been provided on various aspects of the study. However, there are still some points that require further elaboration and/or revision. Specifically:
- The revision of the abstract is mainly limited to phrasing changes (yet, in some cases the meaning is confusing). As previously suggested, a brief presentation of the background of the topic leading to the objectives of the study could be added. Also, it is recommended to avoid using references in the abstract, considering that in many instances the abstract may be used on its own in the article’s description, therefore in such a case the references list will not be available.
- Apart from the provided response in the cover letter, the literature review on the specific topic, the novelty features and, most importantly and the added value related to the previous study (Kwan 2021) should be elaborated in the introduction.
- The known limitations and uncertainties of the study (e.g. based on the existing simplifications / assumptions and data availability) should be summarised in the discussion and conclusions, to clarify to the reader the possible constrains and avoid misinterpretations.
- In some parts of the manuscript the language could be further improved, that would also assist in the smother flow of the text.
